# Motor Coordination in Primary School Students: The Role of Age, Sex, and Physical Activity Participation in Turkey

**DOI:** 10.3390/children10091524

**Published:** 2023-09-08

**Authors:** Tulay Canli, Umut Canli, Cuneyt Taskin, Monira I. Aldhahi

**Affiliations:** 1Institute of Health Sciences, Tekirdag Namık Kemal University, Tekirdag 59010, Turkey; tulayengin018@hotmail.com; 2Sports Science Faculty, Tekirdag Namik Kemal University, Tekirdag 59010, Turkey; 3Kirkpinar Faculty of Sport Sciences, Trakya University, Edirne 22000, Turkey; ctaskin@trakya.edu.tr; 4Department of Rehabilitation Sciences, College of Health and Rehabilitation Sciences, Princess Nourah bint Abdulrahman University, P.O. Box 84428, Riyadh 11671, Saudi Arabia; mialdhahi@pnu.edu.sa

**Keywords:** balance, children, jumping, motor competence, sport participation

## Abstract

Motor coordination (MC) is an essential skill underpinning precise and controlled movements, contributing significantly to daily functioning and overall performance. The developmental trajectory of MC in children is intricately shaped by a spectrum of factors encompassing age, gender, and physical activity engagement. Delving into the complex interrelation of these variables holds the potential to unravel nuanced developmental trends and offer targeted avenues for interventions aimed at augmenting motor proficiency in the pediatric population. This study aimed to assess the differences in MC of primary school students based on sex, age, and physical activity participation (PAP). A total of 848 students from public primary schools, aged between 6–9 years, including 412 boys and 436 girls. The MC was measured using Körperkoordinationstest für Kinder (KTK3+) test battery, which included Jumping sideways (JS), Balancing backward (BB), Moving sideways (MS), and Eye–Hand Coordination (EHC). One-way multivariate analysis of variance (MANOVA) was used to determine the binary and triple interactions of sex, age, and PAP variables on the MC parameters of the participants. The study revealed that boys aged 6–9 had higher scores than girls on eye–hand coordination (EHC) (*p* < 0.02). No significant gender-related differences in balancing backward (BB), jumping sideways (JS), and moving sideways (MS) were found. When the subtests of KTK3+ were compared by age, a significant difference was observed between the groups in all subtests (*p* < 0.05). With respect to PAP, students with PAP had a significant advantage in all subtests of the KTK3+ (*p* < 0.05). The double co-effects or triple co-effects of age, sex, and PAP parameters do not influence the KTK parameters. This study presents evidence supporting sex differences in the motor skills of children within this age range and highlights the potential impact of age and physical activity on motor development.

## 1. Introduction

Motor coordination (MC) plays an important role in human development [1]. It encompasses the harmonious execution of various movements by the nervous system and muscles, reflecting physical fitness [2]. MC constitutes a foundational motor skill and should be recognized as a pivotal component of children’s health-related fitness [3,4]. Optimal MC levels are essential not only for good general growth but also for fostering health, psychological development, and overall well-being [5,6]. In addition to enhancing motor skills, MC can influence cognitive and academic achievements as well as successful engagement in social and physical activities among peers [7,8,9].

There are many factors that affect the level of MC in children. Previous research has identified various biological (e.g., age, sex, body weight), behavioral (e.g., physical activity), socioeconomic (e.g., parents’ economic and educational level), and environmental (e.g., access to sports facilities) factors that affect MC in children [10,11]. The interplay of biological, social, and environmental factors, including sex differences and age-related changes, can significantly shape MC abilities. Understanding this interplay is crucial for providing comprehensive support to promote healthy development in primary school students, especially given that their participation in physical activities is known to have a positive impact on both their health and development [12].

Research has indicated that there are often differences in MC between boys and girls. In general, boys tend to exhibit higher levels of gross motor skills, such as running and throwing, while girls may excel in fine motor skills, such as handwriting and manipulative tasks [13]. A significant difference in locomotor skills based on sex has been reported, with the mean motor quotients (KTK) of boys (83.34) being significantly higher than that of girls (72.39) [14]. However, some studies have found that girls and boys have similar MC levels [15,16]. The disparities in existing research highlight the need for further studies to determine the effect of sex on MC. In addition, age has been shown to affect MC levels [17]. For example, Bolger et al. [18] found that age had a significant effect on MC levels, and Lee et al. [19] demonstrated that as primary school children got older, their MC skills improved. Moreover, several studies on the relationship between age and MC levels in athletes and non-athletes of different ages have revealed that as the age level increases, the level of MC also increases [20,21,22].

A previous study indicated that MC was the best predictor of physical activity levels in children 6 to 10 years of age [3]. By participating in physical activities, children gain more experience and refine their motor skills [23,24]. For example, children can improve their MC skills through various physical activities during recess and participation in physical education classes [25]. Indeed, many studies have identified positive relationships between physical activity levels and MC, particularly in primary school students [26,27]. Other studies have specifically examined the effects of physical activity or sports participation on MC. However, there is a lack of research on the synergistic effects of all these factors on MC, especially in primary school-aged children. Therefore, we aimed to characterize the MC levels of the students and to investigate the association between age, sex, and PAP on MC. The findings of this study could address this prevailing gap in the literature. Moreover, determining the combined effect of all factors on MC can contribute to the development of more appropriate exercise strategies for individuals in childhood.

## 2. Materials and Methods

### 2.1. Study Design and Participants

In this cross-sectional study, a total of 848 students (48.58% boys, *n* = 412) aged between 6–9 years who attend official primary schools were evaluated. Children with any orthopedic, cardiological, or neurological diseases were excluded from the study. Additionally, the participants were instructed to refrain from engaging in strenuous physical activity prior to measurements and to avoid the use of painkillers or sleeping pills one day before the measurements. This protocol was enforced to maintain consistency and accuracy in the measured outcomes. A total of 864 students were initially included in the study. However, 16 students who did not meet the stated conditions for participation were excluded from the study. Thus, in this cross-sectional study, a total of 848 students between six and nine years of age who attend official primary schools were evaluated. The sample size was calculated based on accordance with Cingi’s study [28] and the number of students in the District Directorate of National Education. It indicated an approximate total of 10,000 students attending primary schools in the area. A sample size of at least 370 students was estimated with a margin of error of 5%, a z-score of 1.96, and a confidence interval set to 95%.

### 2.2. The Study Ethical Considerations

The research sought approval from the Tekirdag Namik Kemal University Scientific Research and Publication Ethics Committee. Ethical committee approval was obtained under Protocol No: 2023.53.03.12. To conduct the research in primary schools, an application was made to the Tekirdağ Provincial Directorate of National Education. The necessary permissions were granted by the Tekirdag Provincial Directorate of National Education and Tekirdag Governorship (Tekirdag Provincial Directorate of National Education letter number: 34272485, date: 11 October 2021). Furthermore, a voluntary and parental informed consent form, which included details about the research’s purpose, objective, method, and permission, was sent to student families. Students who provided permission were included in the research.

### 2.3. Procedure

On the first day of data collection, the participants completed a demographic questionnaire, which captured their physical characteristics, and their anthropometric measurements were recorded (e.g., height and weight). The second day of the research process involved explaining and demonstrating the measurement protocols for the KTK3+ battery in detail to the students in accordance with the original testing manual guidelines [29,30]. The sub-tests of the KTK3+ test battery, which included Jumping sideways (JS), Balancing backward (BB), Moving sideways (MS), and Eye–Hand Coordination (EHC), were administered in the same order to all participants in the study by the same researchers. A standard warm-up was performed prior to testing, which involved five minutes of jogging and five minutes of dynamic stretching.

### 2.4. Instruments

#### 2.4.1. Anthropometric Measures

The participants’ height was measured using a portable height device (Mesilife 13539) with a sensitivity of 0.1 cm. During the measurement process, the athletes were instructed to stand upright barefoot with their heads held high, feet flat on the ground, knees straight, heels together, and bodies erect [30]. The participant’s body weight (in kilograms) was measured barefoot while wearing only shorts and a t-shirt using a digital scale (PoloSmart PSC05 Mood Glass Digital Scale) with a sensitivity of 0.01 kg [31]. The body mass index (BMI in kg/m^2^) was determined by dividing their body weight (in kg) by the square of their height.

#### 2.4.2. KTK3+ Test Battery

In this study, the MC levels and movement competencies were determined by applying the KTK3 [32] supplemented with an eye–hand coordination task [33], resulting in the KTK3+ [34]. KTK3 is a measurable (quantitative) test tool with high validity and reliability used to evaluate general gross MC [29,32,34,35]. KTK3 consists of three test elements commonly used on a global scale. The first test is lateral jumping, and participants are required to jump on a wooden platform with both feet for 15 s. The result score is obtained from the total number of skipped jumps during both trials. In the second test, participants are required to move sideways using two wooden platforms for 20 s. The total score is obtained by adding the number of times participants lowered one wooden platform and touched the moving wooden platform during both trials. The third and final test of KTK3 is backward balancing, which can be done on three narrowing balance beams (from 6.0 cm to 4.5 cm and 3.0 cm). The total number of steps is counted, and a maximum of 72 steps (or 8 steps on each balance beam in each trial) can be taken.

#### 2.4.3. Eye–Hand Coordination Test (EHC)

The EHC task used by Platvoet et al. [33] requires an individual to throw a tennis ball against a wall with one hand and catch it with the other hand. This standardized test objectively evaluates an individual’s ball control and capacity for anticipation. The EHC task can be easily administered in different (large) environments and is particularly relevant to sports [30,33]. Platvoet et al. [33] reported a significant increase in raw scores across various age groups, with boys showing higher scores as compared to girls. Although based on limited age range data, these findings clearly demonstrate that the KTK3 test can cover a wide range of general motor performance skills and distinguish between children with different levels of MC when combined with an EHC task. Coppens et al. [34] conducted a study to validate the KTK3+ test battery and provided updated normative MC values for boys and girls between the ages of 6 and 19 years old. Factor analysis with multidimensional scaling showed that the one-dimensional model provided the best fit and that all test items were associated with the same potential structure MC, which was further supported by moderate to good relationships between all four test items (r = 0.453–0.799). 

### 2.5. Statistical Analyses

The data obtained in the research were analyzed using the SPSS 18.0 program. Descriptive statistics (mean, standard deviation, frequency, and percentage values) were used to describe the characteristics of the participants. Skewness and kurtosis values were examined to determine whether the research data were normally distributed. In addition, a three-way multiple analysis of variance (MANOVA) was used. *p*-values less than 0.05 were considered to indicate statistical significance. However, to control for type I errors in all possible multiple comparisons, the Bonferroni correction was applied. In studies involving more than two groups, conducting all possible pairwise comparisons increases the risk of type I errors. The Bonferroni test adjusts the significance level in such cases. The initially accepted significance level (e.g., *p* = 0.05) is divided by the number of possible comparisons (e.g., 3). The calculated F-value is then compared not with the value corresponding to the 0.05 level in the relevant table but with the recalculated *p*-value (0.0166 ≅ 0.01) [36]. Additionally, a Scheffe post hoc analysis was conducted to determine which age groups were responsible for the differences. The construct validity of the scale used was tested using the IBM AMOS 24.0 program through structural equation modeling to determine whether the KTK3+ measured the intended construct of MC on the sample of Turkish children. Second, the correlation between all samples in which the scale was administered was examined using the SPSS 18.0 program, and the internal consistency and reliability of the samples were calculated using Cronbach’s alpha reliability coefficient and Confirmatory Factor Analysis to assess the proposition that a connection between observed variables and their underlying constructs is present.

## 3. Results

The KTK3+ test was found to have a high degree of fit values with all sample groups and all measurement tests using the structural equation modeling (Table 1). The graphics of the models obtained by the structural equation modeling are shown in Figure 1, which confirms the validity of the test among Turkish children.

The Cronbach’s alpha coefficients, which are indicators of internal consistency and reliability, were calculated to determine the reliability of the KTK3+ measurement tool consisting of four tests. The reliability coefficients for the measurement phase (n = 848) of the KTK3+ scale were shown in the following table, including all sample groups and tests, to check the appropriateness of the scale (Figure 1). When the reliability coefficients of the KTK3+ Scale were examined, it was found that the obtained values had very high reliability for all sub-tests of the KTK3+ test for all sample groups. The KTK3+ test provided highly consistent and reliable results for all samples simultaneously, only for girls, only for boys, and for all categories of the 6–9 age group. The findings of the research indicate that the KTK3+ test is a highly valid and reliable measurement tool to be used by Turkish children (6–9 years old, girl, boy) (Table 2).

Table 3 represents the frequency and percentage of sex distribution across the children’s ages. The values of the anthropometric measurements of the participants in terms of sex and age variables are shown in Table 4. Descriptive statistics of the KTK3+ scores of participants by age, sex, and PAP are represented in Table 5.

Boy participants had higher EHC scores than girl participants (*p* < 0.025). No significant differences were found in the BB, JS, and MS variables based on sex (*p* > 0.05). However, all of the KTK variables showed significant differences based on age group (KTK_BB_ F = 32.85; *p* = <0.001; η^2^ = 0.10, KTK_JS_ F = 35.65; *p* = 0.00; η^2^ = 0.11, KTK_MS_ F = 33.83; *p* = <0.001; η^2^ = 0.10, KTK_EHC_ F = 128.00; *p* = <0.001; η^2^ = 0.31). Based on the Scheffe post hoc analysis, the KTK BB, JS, and MS scores increased with age (Table 5). No significant differences were found between the scores of the eight- and nine-year-old age groups in BB, JS, and MS parameters (*p* > 0.05). However, significant differences were observed in all other age groups (*p* < 0.05). EHC increased with age, and there were significant differences between all age groups (*p* < 0.05). After applying the Bonferroni correction for PAP, the *p*-value was calculated as 0.025. Based on this result, all of the KTK variables differed significantly by PAP (KTK_BB_ F = 8.62; *p* = 0.003; η^2^ = 0.01. KTK_JS_ F = 47.92; *p* = 0.00; η^2^ = 0.05. KTK_MS_ F = 7.30; *p* = 0.007; η^2^ = 0.00. KTK_EHC_ F = 10.14; *p* = 0.001; η^2^ = 0.01). The interaction of sex × age, sex × PAP, age × PAP, and sex × age × PAP did not significantly affect any KTK parameter (*p* > 0.05) (Table 6).

## 4. Discussion

The aim of this study was to investigate the influence of sex, age, and physical activity on the MC levels of primary school students. Further, the interactions among sex, age, and physical activity in relation to students’ MC were analyzed. 

In this study, boys had higher EHC scores than girls, and the differences in scores among the groups were significant. No significant differences were found in the BB, JS, and MS variables based on sex. In a study involving a group of 665 students between the ages of five and seven years, it was observed that girls had a notable advantage on the BB sub-test, whereas boys had higher scores on all other sub-tests as well as a higher total MC score [37]. Similarly, another previous study [13] reported that boys have higher MC performance than girls. An examination of the MC levels of tennis players between the ages of 6 and 14 years found no significant differences between boys and girls in KTK raw sub-test scores or total KTK motor coefficient values [16]. The author claimed that with equal opportunities for training, boys and girls in this age range may exhibit similar motor skills. Coppens et al. [34] demonstrated that boys performed systematically better than girls in three out of four KTK3+ test items, while girls were more successful than boys in the balance test. These results are in line with the notion that sex affects general motor abilities in various ways, which can be explained by biological effects on motor development, as previously theorized by Barnett et al. [10].

In our study, it was found that as age increased, scores on the trials also increased. This finding is in line with a study of 1276 Portuguese children aged 6–14 who had their MC measured with the KTK, which also reported an increase in raw scores with age [20]. The KTK test was used to determine the MC characteristics of Portuguese children aged 6–11 years. It was determined that MC scores increased with increasing age in both girls and boys [38,39]. The same results were obtained in the study conducted by Valdivia [40] among Peruvian children aged 6–11. Further, the age-related MC improvement observed in this study is consistent with the studies of Ahnert et al. [41], Vandorpe et al. [22], and Coppens et al. [34], in which older participants outperformed their younger counterparts. The study of Rodrigues et al. [42] and the systematic review of Barnett et al. [10] also reported a positive relationship between age and motor competence. In our study, there were no significant differences observed between the scores of the eight and nine age groups in the parameters of BB, JS, and MS. This suggests that the eight and nine age groups share similar characteristics. The existing literature does not report any evidence suggesting a disparity or resemblance between these particular age groups, especially in regard to MC.

In the current study, students with PAP had higher scores on all sub-tests compared to their peers without PAP, regardless of sex and age group, and there was a statistically significant difference between the groups. Schembri et al. [43] study on primary school students revealed that students who participate in sports organizations have better MCs than students who do not. It has been stated that the results of this study are useful in understanding the role of the school in the development of MC and planning adequate strategies to overcome the current limitations in the physical education teaching-learning process. Further, Sogut [44] compared the MC levels of young tennis players of different performance levels and considered variables such as experience, weekly training volume, and level of competition. The results showed that players in the elite group exhibited statistically higher MC performance. Additionally, early involvement in sports and weekly training volume was suggested to have positive effects on MC. 

Previous studies have shown that physical activity in childhood, including participation in organized sports, has a positive effect on the development of motor competence [45]. Coppens et al. [34] used the KTK3+ test battery to compare participants who engaged in organized sports with those who did not on a weekly basis. Those who were not engaged in organized sports obtained systematically lower scores for all motor tasks and this difference was observed in both young and old age groups. However, the effect size in this study was relatively small [34]. 

In this study, the MANOVA analysis revealed no significant differences in the combined dependent variables for the interaction of sex × age, sex × PAP, age × PAP, and sex × age × PAP. Notably, only one study utilizing the same analysis method and KTK3+ test battery was found in the literature. Coppens et al. [34] showed a tendency for significant sex × age group interaction effects in the JS test and significant effects in the EHC task. Further examination of this interaction revealed that boys outperformed girls in both the JS and EHC tasks in every age group. However, the difference between boys and girls in the JS test was more pronounced in older age groups (>16.00 years). For the EHC task, the sex difference decreased as age increased. Girls outperformed boys in the BB test, while the main effect of sex in the MS test favored boys. Regardless of sex and PAP, a significant increase in scores was observed in all age groups for the four test components of the KTK3+ battery. 

In practice, the results of this study emphasize the importance of participation in physical activity. The most important implication of the study is that MC elements can be developed by including children in organized physical activity programs and the combined effects of age, sex, and PAP. Future studies should examine the long-term effects of sustained physical activity on MC. Longitudinal research can help us understand how continuous participation in physical activities throughout childhood and adolescence impacts MC in adulthood. In addition, examining specific aspects of children’s physical fitness and determining their relationship with MC would provide more information about other factors affecting MC.

This study has some limitations that should be acknowledged. Firstly, the design of the study cannot establish causation underlying the changes in the MC and differences based on age and sex. To gain deeper insights into the development of motor skills, conducting longitudinal and experimental studies is essential. Secondly, to better understand students’ physical activity levels, more detailed information is needed. This includes comprehensive data on participation in organized sports or non-organized sports activities. Furthermore, factors such as family sports history, socioeconomic status, and education level should be considered for a more comprehensive analysis. Lastly, it is crucial to emphasize that this particular study was conducted exclusively in a specific region of Turkey. Consequently, the conclusions reached may not be directly applicable to the broader population of Turkey or other regions that possess distinct demographic, cultural, or socioeconomic characteristics. It is recommended to conduct further research that includes diverse regions in order to obtain a more comprehensive understanding.

## 5. Conclusions

The results of the study reveal that boys obtained higher scores in the EHC (Eyes Half-Closed) test compared to girls. However, there were no statistically significant differences in the variables of BB (Balancing Backward), JS (Jumping Sideways), and MS (Moving Sideways) based on sex. Across all subtests, there was a positive correlation between age and test scores, indicating that as age increased, the test scores also increased. However, when comparing only the 8 and 9-year-old age groups, no statistically significant differences were found in the BB, JS, and MS tests. Moreover, students with high PAP exhibited higher scores in all subtests compared to those who did not participate in physical activity. This suggests a positive association between physical activity levels and motor skills performance among the participants.

## Figures and Tables

**Figure 1 children-10-01524-f001:**
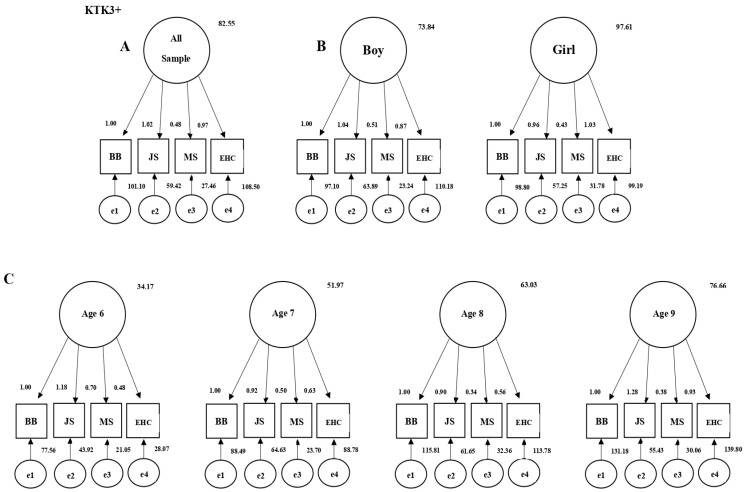
Structural equation models of the KTK3+ test (**A**): All Sample. (**B**): Boy and Girl Sample. (**C**): The 6 to 9 Age Sample).

**Table 1 children-10-01524-t001:** Structural equation model goodness-of-fit indices and normal values.

Fit İndices	Good Fit	Acceptable Fit	All Sample	Boy	Girl	Age 6	Age 7	Age 8	Age 9
x^2^ “*p*”	*p* > 0.05	-	0.37 *	0.29 *	0.10 *	0.62 *	0.11 *	0.22 *	0.47 *
x^2^/df	<3	<5	0.99 *	1.20 *	2.22 *	0.47 *	2.18 *	1.48 *	0.75 *
GFI	>0.95	>0.90	0.99 *	0.99 *	0.99 *	0.99 *	0.98 *	0.99 *	0.99 *
AGFI	>0.95	>0.90	0.99 *	0.98 *	0.97 *	0.99 *	0.94 *	0.96 *	0.98 *
CFI	>0.95	>0.90	1.00 *	0.99 *	0.99 *	1.00 *	0.97 *	0.98 *	1.00 *
RMSEA	<0.05	<0.10	0.00 *	0.02 *	0.05 *	0.00 *	0.07 *	0.04 *	0.00 *
RMR	<0.05	<0.10	1.14	1.52	2.31	0.76	2.84	3.36	2.69

* = the fit index values.

**Table 2 children-10-01524-t002:** Cronbach’s alpha reliability coefficients of KTK3+ scale.

KTK3+ Test	All Sample	Boy	Girl	Age 6	Age 7	Age 8	Age 9
Balancing backwards	0.86	0.85	0.88	0.81	0.81	0.81	0.84
Jumping sideways	0.88	0.87	0.89	0.85	0.82	0.82	0.88
Move sideways	0.77	0.78	0.75	0.82	0.76	0.65	0.67
EHC	0.85	0.83	0.88	0.73	0.76	0.75	0.83

**Table 3 children-10-01524-t003:** Description of the age and sex of the study participants.

Variables		Age Group								Total	
		6 Years		7 Years		8 Years		9 Years			
		N	%	N	%	N	%	N	%	N	%
Sex	Boy	102	12.03	100	11.79	107	12.61	103	12.14	412	48.57
	Girl	135	15.91	100	11.79	100	11.79	101	11.94	436	51.43
Total		237	27.94	200	23.58	207	24.40	204	24.08	848	100

**Table 4 children-10-01524-t004:** Descriptive statistics of the physical characteristics of participants by sex and age.

Variables	Sex	N	Mean ± Sd	Age	N	Mean ± Sd
Height (cm)	Boy	412	129.02 ± 7.79	6 Years	237	121.12 ± 5.59
				7 Years	200	126.11 ± 5.63
	Girl	436	128.13 ± 8.76	8 Years	207	131.66 ± 6.46
				9 Years	204	136.46 ± 5.99
Weight (kg)	Boy	412	29.84 ± 8.65	6 Years	237	24.12 ± 5.83
				7 Years	200	27.85 ± 6.91
	Girl	436	29.42 ± 8.69	8 Years	207	31.95 ± 8.31
				9 Years	204	35.41 ± 8.93
BMI (kg/m^2^)	Boy	412	17.81 ± 3.46	6 Years	237	16.51 ± 2.72
				7 Years	200	17.45 ± 3.08
	Girl	436	17.63 ± 3.42	8 Years	207	18.22 ± 3.55
				9 Years	204	18.87 ± 3.91

Sd = Standard deviation; BMI = Body Mass Index.

**Table 5 children-10-01524-t005:** Descriptive statistics of the KTK3+ scores of participants by age, sex, and PAP.

Variables	Age Groups	Mean ± Sd	Groups	Mean ± Sd	Groups	Mean ± Sd
KTK_BB_	6 Years	23.04 ± 0.90	Boy	28.96 ± 0.70	With PAP	31.40 ± 0.85
	7 Years	28.51 ± 1.06				
	8 Years	32.60 ± 1.02	Girl	30.93 ± 0.70	Without PAP	28.49 ± 0.50
	9 Years	35.63 ± 0.92				
KTK_JS_	6 Years	31.90 ± 0.82	Boy	38.47 ± 0.60	With PAP	41.15 ± 0.73
	7 Years	36.93 ± 0.92				
	8 Years	40.71 ± 0.87	Girl	37.90 ± 0.60	Without PAP	35.22 ± 0.43
	9 Years	43.18 ± 0.79				
KTK_MS_	6 Years	25.15 ± 0.47	Boy	29.19 ± 0.34	With PAP	29.35 ± 0.42
	7 Years	28.35 ± 0.52				
	8 Years	29.60 ± 0.50	Girl	28.18 ± 0.34	Without PAP	28.02 ± 0.25
	9 Years	31.64 ± 0.45				
KTK_EHC_	6 Years	3.82 ± 0.82	Boy	16.73 ± 0.60	With PAP	16.70 ± 0.73
	7 Years	12.32 ± 0.92				
	8 Years	20.13 ± 0.88	Girl	13.95 ± 0.60	Without PAP	13.97 ± 0.43
	9 Years	25.08 ± 0.79				

**Table 6 children-10-01524-t006:** Description of the raw scores for each KTK3+ test by age category and PAP.

Variables	Sex			Age			PAP			Sex × Age			Sex × PAP			Age × PAP			Sex × Age × PAP		
	F	*p*-Value	η^2^	F	*p*-Value	η^2^	F	*p*-Value	η^2^	F	*p*-Value	η^2^	F	*p*-Value	η^2^	F	*p*-Value	η^2^	F	*p*-Value	η^2^
Multivariate	6.37	<0.001	0.03	26.78	0.00	0.11	12.21	0.00	0.05	1.17	0.29	-	0.57	0.68	-	0.97	0.47	-	0.38	0.97	-
KTK_BB_	3.96	0.04	0.00	32.85	0.00	0.10	8.62	0.00	0.01	1.85	0.13	-	0.24	0.61	-	0.69	0.55	-	0.29	0.82	-
KTK_JS_	0.44	0.50	-	35.65	0.00	0.11	47.92	0.00	0.05	0.64	0.58	-	0.02	0.88	-	0.77	0.50	-	0.32	0.80	-
KTK_MS_	4.30	0.03	0.00	33.83	0.00	0.10	7.30	0.00	0.00	1.21	0.30	-	0.26	0.60	-	3.10	0.02	-	0.14	0.93	-
KTK_EHC_	10.52	0.00	0.01	128.00	0.00	0.31	10.14	0.00	0.01	0.01	0.99	-	1.14	0.28	-	1.14	0.28	-	0.31	0.81	-

BB = Balancing backward; JS = Jumping sideways; MS = Moving sideways; EHC = Eye–hand coordination.

## Data Availability

Not applicable.

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
