# Peer review of "Motor Coordination in Primary School Students: The Role of Age, Sex, and Physical Activity Participation in Turkey"

_children, 2023, doi:10.3390/children10091524_

Round 1
Reviewer 1 Report
Dear Editor,
thank you for the opportunity to review this manuscript. The work is good, being data collected on a large sample of 848 students. However, the manuscript needs some adjustments, both general and specific, as follows.
From the abstract, the research problem cannot be deduced.
In keywords' section you should avoid inserting words (e.g. coordination) already present in the text.
You should expand the background of the study.
In the results' section, before each table, you should include the corresponding description. Now it is very confusing. Furthermore, you argued the results of the post hoc bonferroni, which you did not mention in the 'statistical analysis' section of the method. You should enter step by step all the statistical tools used.
What do the results mean in practice? What is the usefulness of the study? What are the future implications?
Generally, you should revise:
- punctuation and writing. For example in line 103 you repeat twice 'On the second day'
- english language
- the style of references (some of them are even missing). For example, in citation n. 25 you should remove '(2012)':
Kalaja, S.P.; Jaakkola, T.T.; Liukkonen, J.O.; Digelidis, N. (2012). Development of middle school students' fundamental movement skills of movement and physical activity in a naturalistic physical education context. Pedagogy of physical education and sport. 2012, 17(4), 411-428.
Moderate editing of English language required.
Author Response
Dear Reviewer 1, Hi
I showed all changes in the manuscript. Also, I am sharing answers to your comments and suggestions in a different file.
Best regards.

Reviewer 2 Report
First of all, I would like to thank you for the opportunity to evaluate the manuscript whose aim has been to assess the differences in motor coordination (MC) of primary school students based on sex, age, and physical activity participation (PAP), as well as investigation the binary and triple interactions of these variables on MC.
Title: It is correct although it must be indicated where the study has been carried out (country or region)
Abstract: The abstract should be improved. At the beginning, according to the rules of the journal, a small background must be included. (Authors should take into account that they should not exceed the maximum number of words). The results of lines 20-21 are not correct. Review all the results included according to gender and sports practice
Key words: They must be improved. For example authors are encouraged to change youth to children or include KTK3
Introduction: Although it is easy to follow, it should be improved because the purpose of the study is not reflected at the end of it. The reference11 refers to adolescents. Authors are encouraged to include appropriate references to the study.
The paragraph from lines 70 to 79 should not be written there. Authors are recommended to move to the conclusions section highlighting the importance of this study.
Authors should include at the end of the introduction the research questions that arise from what is stated in the introduction itself and include the objective or objectives of the study.
Methods:
Study design and participants: Authors are requested to include the calculation of representativeness of the sample used.
How many students have been dismissed for not meeting the exclusion criteria? this data must be reported in the document.
Line 89 -99: They are ethics, therefore it should be in a section that is identified with that title.
Line 106-107: Before using these abbreviations, the tests must be defined.
Line 109: Authors should include if and where tests were developed individually. The authors must remember that the object of the investigation is that it can be replicated in the same circumstances.
Statistical analyses: line 159: Authors are requested to indicate and include the homogeneity analysis. What post hoc tests have been used? What has been the confidence index and the statistical significance used? Has the statistical power been calculated?
Results.
This section is the one suggested to the authors who should improve it so that this article can be evaluated for publication.
Line 167-169. It is not understood why the authors begin the results with Tables 3 and 4 when Table 1 appears after that paragraph. In addition, the results must identify the characteristics of the sample (i.e. number of boys and girls, ages, participating schools, weight and height, BMI, etc.)
Line 170. table 1, figure 1, table two, must be referenced in the text of the article
Line 198. Table 5 should report the means and Standard Deviations obtained by the participants in each of the KTK3 tests. In the written text, it should be reported whether or not these significant differences exist. Reviewers cannot infer the results of the tests.
Line 200: The results have not been described correctly. Authors should refer to the table or figure to which they refer when presenting the results.
The authors have not reported whether there is a significant main effect of age, gender or physical activity practiced or the interaction between these factors. The authors are requested to include these results.
Line 201-202: The authors indicate that no statistically significant differences have been found in these dimensions of the KTK3. How is this possible without Table 5 if these significant differences appear (BB (p = 0.047); MS (p = 0.038; EHC P = 0.001)? Authors should review all these results and rewrite this section.
Line 207; This post hoc analysis has not been reported in the statistical analysis section. Why has Scheffe been chosen and not Bonferroni?
Discussion:
Authors are encouraged to redo this section once they have improved and analyzed the study results as previously proposed.
The discussion must be rewritten based on the contributions that have been made in the results section by this reviewer.
Line 226. These results are not understood. If it is another study, the authors must report the appointment. If it is not another study, the number of students does not coincide with the sample reported by the authors in the corresponding section.
Conclusions:
They are not consistent with the results
Line 295: The limitations of the study should go at the end of the discussion and not at the end of the conclusions. Furthermore, this aspect should be improved.
References:
Authors are suggested to improve the references. Of the 38 references, only 12 (31%) are from the last 5 years. This aspect should be improved.
I hope these comments help the authors to improve their article.
kind regards
Author Response
Dear Reviewer 2, Hi.
I showed all changes in the manuscript. Also, I am sharing the answers to comments and suggestions in a different file.
Best regards.

Round 2
Reviewer 1 Report
Dear Editor,
the manuscript has changed significantly from the previous version. I recommend re-reading it to correct some minor superficial errors. For example, the word 'motor coordination' must be abbreviated the first time it is mentioned in MC (and not the second). Thereafter, ONLY the abbreviation must be repeated. Also 'Physical Activity Participation' which was initially abbreviated to PAP, is then repeated in the text in full.
Author Response
Dear Reviewer,
We greatly appreciate the time and effort you dedicated to reviewing our work. We are pleased to inform you that we have carefully considered your feedback and have made significant revisions to the paper based on your suggestions. We also send the paper for proofreading to enhance the English writing. Your comments have played a crucial role in improving the clarity, coherence, and overall quality of our research. Your expertise and thoughtful critique have been invaluable in identifying areas that required further development and refining our methodology. We have diligently ensured terminological consistency throughout the manuscript.
Thank you for your valuable support and guidance throughout the review process.

Reviewer 2 Report
Tittle: Although the authors say that they do not want to indicate the region, It must be reported for the interest of the research. That is to say, it is not the same that a batting test is applied in Anglo-Saxon countries where baseball is a major sport than it is done in another in which this sport does not have much incidence. For these reasons, the region where the study was carried out should be included. Also, I do not understand the response of the authors since in line 159 they indicate that it is a study in the primary schools of Suleymanpasa (Turkey).In addition, depending on the number of subjects participating in the study, this may or may not be generalized to a specific place, region or country. Line 27-28: Delete (p= 0.294; 0.680; 0.471; 0.970) Introduction: Line 38: “MC” should be moved to line 36 after Motor coordination and from there always use MC Method. Study desing and sample size: What is the margin of error (1%-3%-5%) that the authors have taken into account in calculating the sample size? It is not reported in the document. Results: This test is validated. I don't know if it is for the Turkish population, but in any case, it would be worth performing a confirmatory factor analysis of the instrument and reporting the results of the goodness of fit indices (i.e. root mean square standardized residual (SRMR), root mean square error of approximation (RMSEA ), non-regulated fit index (NNFI)…). Line 251-252: the age of the participants should be a covariate since according to the existing scientific literature, it is known that motor competence increases with increasing age, simply due to the maturation of the subject. Table 1 and 2 and figure 1 are not cited in the text of the manuscript. Authors are invited to review this and cite tables and figures from the document in the text as indicated in the previous review.
Line 291: it seems that a table number is missing. Is this possible?
Author Response

(The authors gave the same response as above.)
